# PCR-Based Screening of Spinal Muscular Atrophy for Newborn Infants in Hyogo Prefecture, Japan

**DOI:** 10.3390/genes13112110

**Published:** 2022-11-14

**Authors:** Yoriko Noguchi, Ryosuke Bo, Hisahide Nishio, Hisayuki Matsumoto, Keiji Matsui, Yoshihiko Yano, Masami Sugawara, Go Ueda, Yogik Onky Silvana Wijaya, Emma Tabe Eko Niba, Masakazu Shinohara, Yoshihiro Bouike, Atsuko Takeuchi, Kentaro Okamoto, Toshio Saito, Hideki Shimomura, Tomoko Lee, Yasuhiro Takeshima, Kazumoto Iijima, Kandai Nozu, Hiroyuki Awano

**Affiliations:** 1Department of Clinical Laboratory, Kobe University Hospital, 7-5-1 Kusunoki-cho, Chuo-ku, Kobe 650-0017, Japan; ynoguchi@med.kobe-u.ac.jp (Y.N.); hisamt@med.kobe-u.ac.jp (H.M.); kmatsui@med.kobe-u.ac.jp (K.M.); yanoyo@med.kobe-u.ac.jp (Y.Y.); 2Department of Pediatrics, Kobe University Graduate School of Medicine, 7-5-1 Kusunoki-cho, Chuo-ku, Kobe 650-0017, Japan; ryobo@med.kobe-u.ac.jp (R.B.); iijima@med.kobe-u.ac.jp (K.I.); nozu@med.kobe-u.ac.jp (K.N.); awahiro@med.kobe-u.ac.jp (H.A.); 3Department of Community Medicine and Social Healthcare Science, Kobe University Graduate School of Medicine, 7-5-1 Kusunoki-cho, Chuo-ku, Kobe 650-0017, Japan; yogik.onky.s@mail.ugm.ac.id (Y.O.S.W.); niba@juntendo.ac.jp (E.T.E.N.); mashino@med.kobe-u.ac.jp (M.S.); 4Department of Occupational Therapy, Faculty of Rehabilitation, Kobe Gakuin University, 518 Arise, Ikawadani-cho, Nishi-ku, Kobe 651-2180, Japan; 5Department of Pediatrics, Kobe City Medical Center General Hospital, 2-1-1 Minatojimaminami-machi, Chuo-ku, Kobe 650-0047, Japan; msm.sugawar@gmail.com; 6Department of Pediatrics, Ueda Hospital, 1-1-4 Kunikadori, Chuo-ku, Kobe 651-0066, Japan; go_ueda_go@yahoo.co.jp; 7Department of Biochemistry, Faculty of Medicine, Public Health, and Nursing, Universitas Gadjah Mada, Jalan Farmako, Yogyakarta 55281, Indonesia; 8Laboratory of Molecular and Biochemical Research, Biomedical Research Core Facilities, Juntendo University Graduate School of Medicine, Hongo, Bunkyo-ku, Tokyo 113-8421, Japan; 9Faculty of Nutrition, Kobe Gakuin University, 518 Arise, Ikawadani-cho, Nishi-ku, Kobe 651-2180, Japan; bouike@nutr.kobegakuin.ac.jp; 10Instrumental Analysis Center, Kobe Pharmaceutical University, 4-19-1 Motoyamakitamachi, Higashinada-ku, Kobe 658-8558, Japan; takeuchi@kobepharma-u.ac.jp; 11Department of Pediatrics, Ehime Prefectural Imabari Hospital, 4-5-5 Ishi-cho, Imabari 794-0006, Japan; kentaro206@gmail.com; 12Department of Neurology, National Hospital Organization Osaka Toneyama Medical Center, 5-1-1 Toneyama, Toyonaka 560-8552, Japan; saito.toshio.cq@mail.hosp.go.jp; 13Department of Pediatrics, Hyogo Medical University, 1-1 Mukogawacho, Nishinomiya 663-8501, Japan; shimomura.ped@gmail.com (H.S.); leeleetomo@me.com (T.L.); ytake@hyo-med.ac.jp (Y.T.); 14Hyogo Prefectural Kobe Children’s Hospital, 1-6-7 Minatojimaminami-machi, Chuo-ku, Kobe 650-0047, Japan

**Keywords:** spinal muscular atrophy, *SMN1* deletion, newborn screening, false positive, heparinized blood, diluted blood, pre-symptomatic treatment, post-symptomatic treatment

## Abstract

Spinal muscular atrophy (SMA) is a common devastating neuromuscular disorder, usually involving homozygous deletion of the *SMN1* gene. Newly developed drugs can improve the motor functions of infants with SMA when treated in the early stage. To ensure early diagnosis, newborn screening for SMA (SMA-NBS) via PCR-based genetic testing with dried blood spots (DBSs) has been spreading throughout Japan. In Hyogo Prefecture, we performed a pilot study of SMA-NBS to assess newborn infants who underwent routine newborn metabolic screening between February 2021 and August 2022. Hyogo Prefecture has ~40,000 live births per year and the estimated incidence of SMA is 1 in 20,000–25,000 based on genetic testing of symptomatic patients with SMA. Here, we screened 8336 newborns and 12 screen-positive cases were detected by real-time PCR assay. Multiplex ligation-dependent probe amplification assay excluded ten false positives and identified two patients. These false positives might be related to the use of heparinized and/or diluted blood in the DBS sample. Both patients carried two copies of *SMN2*, one was asymptomatic and the other was symptomatic at the time of diagnosis. SMA-NBS enables us to prevent delayed diagnosis of SMA, even if it does not always allow treatment in the pre-symptomatic stage.

## 1. Introduction

Spinal muscular atrophy (SMA) is a common autosomal recessive disorder that causes degeneration of anterior horn cells in the human spinal cord and subsequent loss of motor neurons [1]. The median incidence of SMA in Europe has been reported as 11.9 per 100,000 live births (range, 6.3–26.7 per 100,000 live births) [2].

SMA is clinically divided into five subtypes [3]: type 0 (the most severe form with onset in the prenatal period; severe respiratory problems after birth), type I (Werdnig–Hoffmann disease; a severe form with onset before 6 months of age; the inability to sit unsupported), type II (Dubowitz disease; an intermediate form with onset before 18 months of age; the ability to sit unaided, but not to stand or walk), type III (Kugelberg–Welander disease; a mild form with onset after 18 months of age; the ability to stand and walk unaided), and type IV (the mildest form with onset after 30 years of age). Among these subtypes, SMA type I accounts for half of all SMA patients [4] and is a leading inherited cause of infant mortality [5]. Many patients with SMA type I die of respiratory insufficiency by the age of 2 when respiratory support is not available [6]. Meanwhile, patients with types II, III, and IV are able to survive for longer times with limited motor function [3,7].

In 1995, two SMA-related genes, *SMN1* and *SMN2* (survival motor neuron 1 and 2), were identified on chromosome 5q13 [8]. They are highly homologous genes; there is only a single nucleotide difference at nucleotide position 6 between the coding regions of *SMN1* and *SMN2*: 6C in *SMN1* and 6T in *SMN2* [8]. Now, *SMN1* is considered to be a causative gene for SMA, with *SMN2* as a modifying factor of the SMA phenotype. More than 90% of SMA patients are homozygous for *SMN1* deletion, while the rest are compound heterozygotes with a deleted *SMN1* allele and a mutated *SMN1* allele [8]. There is an inverse relationship between *SMN2* copy number and disease severity; that is, a higher copy number of *SMN2* is related to a milder SMA phenotype [9,10]. After discovery of *SMN1* and *SMN2*, an *SMN1* deletion test and *SMN2* copy number analysis were used for SMA diagnosis and prediction of prognosis. Molecular genetic testing, as a procedure of definitive diagnosis for SMA, has replaced the invasive diagnostic procedure of muscle biopsy [3]. Patients benefit from molecular genetic testing as it only requires peripheral blood.

SMA was thought to be incurable. However, clinical trials with nusinersen, onasemnogene abeparvovec, and risdiplam showed that these drugs increased the number of patients with SMA type I that were able to sit unsupported and improved their event-free survival period [11,12,13]. The United States Food and Drug Administration approved nusinersen as the first drug for SMA in 2016, onasemnogene abeparvovec as the second drug in 2019, and risdiplam as the third drug in 2020. The Japanese Ministry of Health, Labour, and Welfare also approved these drugs one year after the United States. Recent clinical trials with nusinersen and onasemnogene abeparvovec targeting pre-symptomatic infants showed that extremely early treatment resulted in normal or sub-normal function in SMA patients that were expected to develop SMA type I or II. These patients became able to stand and walk [14,15,16]. The introduction of new drug treatments could significantly change the disease trajectory from the known natural course of SMA [17].

Thus, newborn screening (NBS) for SMA is now being implemented to maximize the effectiveness of these newly developed drugs. As more than 90% of SMA patients are homozygous for *SMN1* deletion, as mentioned above, the presence or absence of *SMN1* is a good marker for SMA screening. To ensure early diagnosis of SMA caused by homozygous deletion of *SMN1*, NBS for SMA (SMA-NBS) via PCR-based genetic testing with dried blood spots (DBS) on filter paper has been implemented in some countries. Based on a global overview of the current situation of SMA-NBS reported in 2021 [18], SMA-NBS programs have so far detected 288 newborns with SMA out of 3,674,277 newborns screened. The annual proportion of newborns screened for SMA in coming years is expected to increase steadily [18].

In Japan, a prefectural SMA-NBS program was first implemented in Chiba Prefecture in May 2020. The first case was identified by SMA-NBS in Kumamoto Prefecture in April 2021, and this infant was treated with onasemnogene abeparvovec at day 42 after birth [19]. Hyogo Prefecture, in the Kansai region of Honshu Island, Japan, also started a pilot SMA-NBS study in February 2021.

In this article, we provide information about the incidence of SMA in Hyogo Prefecture, Japan, before the start of SMA-NBS. This also contains data on aborted fetuses with SMA. We then describe 12 screen-positive cases in the pilot study of SMA-NBS in Hyogo Prefecture. The 12 screen-positive cases included ten false positives and two confirmed patients. The false positives might be related to heparinized and/or diluted blood in the DBS sample. Both patients carried two copies of *SMN2*; one was asymptomatic and the other was symptomatic at the time of diagnosis.

## 2. Materials and Methods

### 2.1. Patients and Fetuses with SMA

We performed genetic testing for SMA in 515 Japanese patients who were referred to the Department of Community Medicine and Social Healthcare Science, Kobe University Graduate School of Medicine, during the period from 1996 to 2019. These patients were suspected of having SMA or another lower motor neuron disease. In the database of 515 patients, we found 18 SMA patients who were born in Hyogo Prefecture during the period 2007–2016. We also found five fetuses who were diagnosed with SMA during the same period who were not included in the patient database. In addition to these infants and fetuses, we describe herein 12 screen-positive cases identified by SMA-NBS.

Prior to genetic analysis, written informed consent was obtained from the patients or their parents/guardians. All procedures were reviewed and approved by the Ethics Committee of Kobe University Graduate School of Medicine (reference number 1089 and B200086, approved on 5 October 2018 and 24 June 2020, respectively) and were performed in accordance with the ethical standards laid down in the Declaration of Helsinki.

### 2.2. Implementation of SMA-NBS in Hyogo Prefecture

In Hyogo Prefecture, the SMA-NBS program was initiated as a pilot study on 1 February 2021 and is still ongoing. This article is an interim report using data collected from 8336 newborns by 31 August 2022. The pilot study was approved by the Institutional Review Board of Kobe University Graduate School of Medicine (reference number B200086, approved on 24 June 2020).

After obtaining written informed consent from the parents, blood samples were collected from the infants within days 4–6 after birth using a heel-prick procedure in the hospital maternity clinics or obstetric departments. The blood was spotted on filter paper (Toyo Roshi Kaisha, Ltd., Tokyo, Japan) and dried for at least 4 h at room temperature. The dried blood samples were sent to the SMCL Center (Sekisui Medical Creative Laboratory Center; Sekisui Medical Co., Ltd., Ibaraki, Japan) for analysis. The analytical methods used for SMA-NBS and the definitive diagnosis of SMA are described below in Section 2.3 and Section 2.4.

The parents were notified of a positive SMA screening result by an obstetrician or neonatologist and were instructed to take the infant to pediatricians, who would provide consultation and examination of the infant.

### 2.3. SMN1 Deletion Screening with Dried Blood Spots

SMA screening of DBS samples was performed with a SMA screening kit, NeoSMAAT SMN1 (Sekisui Medical Co., Ltd., Tokyo, Japan), in the SMCL Center. Our SMA screening methods included a real-time quantitative polymerase chain reaction (qPCR) assay with fluorescent hybridizing probes, which detects the presence or absence of *SMN1* exon 7. In our procedures, the DNA extraction step from the DBS was skipped: one 1.5 mm DBS punch was placed directly into a PCR tube. These PCR tubes were loaded into a thermal cycler, the LightCycler^®^ 96 system (Roche Applied Science, Mannheim, Germany). Newborns with one allelic deletion of exon 7 on *SMN1* (i.e., compound heterozygotes or carriers) were thus not captured by the screening program.

In the second-tier test for samples with a positive result (or an absence of *SMN1* exon 7) in the first-tier test, DNA extracted with a QIAamp DNA Blood Mini Kit (Qiagen, Hilden, Germany) was used for gene analysis by the NeoSMAAT SMN1 kit. If both the first-tier and second-tier tests of the sample demonstrated positive results, then the infant was reported as a screen-positive case to the medical doctors (neonatologists or obstetricians) in the hospital.

### 2.4. Definitive Diagnosis with Freshly Collected Blood

Definitive diagnosis was made based on multiplex ligation-dependent probe amplification (MLPA) and droplet digital PCR (ddPCR) analyses of freshly collected blood at the Department of Clinical Laboratory, Kobe University Hospital. The MLPA analysis was performed using SALSA MLPA Probemix P021 SMA (MRC-Holland, Amsterdam, The Netherlands), according to the manufacturer’s instructions. MLPA data were analyzed using Coffalyser.Net (MRC-Holland). The ddPCR analysis was performed using a ddPCR™ *SMN1* Copy Number Determination Kit and ddPCR™ *SMN2* Copy Number Determination Kit (Bio-Rad Laboratories, Inc., Hercules, CA, USA), according to the manufacturer’s instructions.

### 2.5. Inspection of Inhibitory Effect of Heparin on PCR Assay with DBS

To test the effects of white blood cell count, gene sequence, and heparin on the PCR amplification efficiency, we prepared DBS samples with heparinized and non-heparinized blood. Heparin was added to the freshly collected blood to make samples with final concentrations of 0, 0.05, and 0.5 units/mL. The characteristics of the blood are shown in Appendix A.

Then, we tested the amplification efficiencies of two genes, *SMN* and the cystic fibrosis transmembrane conductance regulator *(CFTR)* gene exon 4, from DBS samples with heparinized blood. The primers for *SMN* were R111 and 541C770 [8], and the primers for *CFTR* were CF621F and CF621R [20]. For duplex amplification of *SMN* and *CFTR*, a 1.2 mm DBS punch was directly added to 20 µL of PCR reaction mixture including four primers, R111, 541C770, CF621F, and CF621R. The thermal cycler used was GeneAtras (ASTEC Co., Ltd., Fukuoka, Japan). The primer sequences, reagents in the PCR mixture, and PCR conditions used in these experiments are shown in Appendix A, respectively.

Gel electrophoresis of the amplified products was performed on 4% agarose gels and visualized with ethidium bromide. The band intensity was determined using ImageJ [21].

### 2.6. Statistical Analysis

The incidence of SMA was defined as the number of newborn infants with the disease in a year in the population and was expressed as the number of affected infants per 100,000 live births. The 95% confidence intervals of the incidence were calculated based on the Poisson distribution using Microsoft Excel. Population data were provided by the Statistics Bureau, Ministry of Internal Affairs and Communications of Japan.

## 3. Results

Our study consists of three parts: (1) prediction of the number of SMA patients identified by SMA-NBS; (2) pitfalls of SMA-NBS; and (3) patients identified by SMA-NBS (Table 1). The first part includes the incidence studies of SMA before SMA-NBS in Hyogo Prefecture, which is described in Section 3.1 and Section 3.2. The second part includes false-positive case studies of SMA-NBS in Hyogo Prefecture, which are described in Section 3.3 and Section 3.4. The third part is a case report of patients who received pre-symptomatic or post-symptomatic treatment, which is described in Section 3.5 and Section 3.6.

### 3.1. Births of SMA Infants in Hyogo Prefecture during the Period 2007–2016

In Japan, newborn screening is implemented on a prefecture basis. Although there are some reports about the incidence of SMA in Japan, it is unknown whether regional differences exist in the incidence of SMA. Therefore, to validate the number of SMA cases detected by the SMA-NBS programs in Hyogo Prefecture, it is essential to estimate the incidence of SMA based on data from this prefecture.

In this study, we estimated the incidence of SMA using the number of SMA-affected infants that were born or SMA-affected fetuses that were aborted in Hyogo Prefecture from 2007 to 2016 and diagnosed with SMA by genetic testing within this period (Table 2). We used the annual population data provided by the Statistics Bureau, Ministry of Internal Affairs and Communications of Japan, which states that Hyogo Prefecture has a population of ~5.5 million and a total of ~40,000 births per year.

According to Table 2, Hyogo Prefecture is expected to have two or three SMA patients of all types combined each year and one or two patients each year of SMA type I.

### 3.2. Estimated Incidence of SMA in Hyogo Prefecture

During the period 2007–2016, the numbers of newborn infants with SMA (including types I, II, and III) and live births were 18 and 464,149, respectively (Table 2). We thus estimated that the incidence of SMA was 3.88 per 100,000 live births [95% CI: 0.02, 7.74], that is, ~1 in 25,000. We also had information on aborted fetuses with a prenatal diagnosis of SMA, although the numbers of aborted fetuses without a prenatal diagnosis of SMA are unknown. When the number of aborted fetuses with a prenatal diagnosis (5 cases) was added to the number of newborn infants with SMA, the estimated incidence of SMA was 4.96 per 100,000 live births [95% CI: 0.59, 9.32], that is, ~1 in 20,000.

The numbers of newborn infants with SMA type I and live births were 9 and 464,149, respectively (Table 2). We thus calculated that the estimated incidence of SMA type I was 1.94 per 100,000 live births [95% CI: −0.79, 4.67], that is, ~1 in 50,000. When the number of aborted fetuses with a prenatal diagnosis of SMA type I (four cases) was added to the number of newborn infants with SMA type I, the estimated incidence of SMA type I was 2.80 per 100,000 live births [95% CI: −0.48, 6.08], that is, 1 in ~36,000. Here, the diagnosis of SMA subtype in the aborted fetuses was based on the clinical subtype of patients in the same family and/or the copy number of *SMN2*. It should be noted that four out of five aborted fetuses with prenatal genetic testing had SMA type I. The parents of a pre-existing child with SMA type I were unable to continue a pregnancy without prenatal diagnosis because they were concerned about the difficulty of raising two children with SMA.

### 3.3. Frequency of False-Positive Cases

We screened 8336 newborn infants for SMA and found 10 false-positive cases during the period of February 2021 to August 2022 (false-positive rate: 0.12%). Table 3 summarizes the false-positive cases. All of these false-positive cases were newborn infants who were born in core hospitals established in the east, central, and north areas of Hyogo Prefecture. These core hospitals provide medical care for newborns whose survival is in danger, where continuous intravenous infusion and/or heparinization of the blood are basic techniques in the management of high-risk and sick infants.

Seven false-positive cases (cases 1–7) were detected among 4050 infants screened in the first ten months (false-positive rate: 0.17%). The frequency of false-positive cases transiently decreased after asking obstetricians and neonatologists not to use heparinized blood when preparing SMA-NBS samples. However, despite this request, heparinized blood samples were prepared for SMA-NBS after May 2022, causing further false positives (cases 8–10). This suggests that heparinized blood is involved in generating false positives in SMA-NBS.

### 3.4. Influencing Factors for PCR-Based Screening

To clarify the factors causing false positives, we tested the effects of white blood cell count, gene sequence, and heparin on PCR amplification efficiency. In this PCR experiment, we used BIOTAQ^TM^ DNA polymerase and Ampdirect^TM^ buffer, which are the main components of the SMA screening kit.

Here, we performed duplex amplification of *SMN* and *CFTR* using DBS samples prepared using fresh blood from a healthy control (Appendix A). The PCR target sequence of *SMN* was a 202 bp fragment containing an *SMN1*-specific nucleotide (c.840C) and a *SMN2*-specific nucleotide (c.840T). The PCR target sequence of *CFTR* was a 237 bp fragment used as a reference.

We obtained three results from these experiments. First, the PCR product amounts of *SMN* and *CFTR* reflected the conditions of non-dilution (white blood cell count 6700/μL) and two-times dilution (white blood cell count 3350/μL) (Figure 1). This suggests that DBS samples that are poor in white blood cells would be unable to obtain the required amount of PCR products for SMA-NBS.

Second, the PCR product amounts of *SMN* were much smaller than those of *CFTR* (Figure 1). Notably, this experiment was conducted with a high concentration of primers for *SMN*, indicating that the amplification efficiency of *SMN* is much lower than that of *CFTR*. This suggests that *SMN* may be difficult to amplify under some conditions, although *CFTR* may be amplified in those same conditions.

Third, the PCR product amounts of *SMN* and *CFTR* decreased in the presence of heparin (Figure 1). The decrease in PCR product amounts may occur in a dose-dependent manner. This suggests that heparin in the DBS sample may hamper PCR amplification in SMA-NBS.

If the DBS samples are poor in white blood cells and/or contain heparin, the SMA-NBS results may be “indeterminate” or “false-positive”. A lack of PCR amplification of *SMN1* and the reference gene may produce indeterminate results, and a lack of PCR amplification in *SMN1* alone and not the reference gene may produce false-positive results.

### 3.5. Patient 1: A Symptomatic Patient Identified by SMA-NBS

Of the 8336 newborns screened in this study, 12 screen-positive cases were detected by real-time PCR assay. The MLPA assay excluded ten false positives and identified two patients (Table 4, Figure 2 and Figure 3).

Patient 1 was a 6-month-old female infant who was diagnosed with symptomatic SMA type I. She was born in April 2022. She was the first child of non-consanguineous healthy parents. The parents had no family histories of neuromuscular disorders. Her clinical information related to SMA diagnosis and treatments is summarized in Table 4 and Figure 2 and Figure 3.

During pregnancy, there were no notable episodes in the mother and infant. However, after the expected date of birth, induced vacuum-assisted vaginal delivery was needed because of severe variable deceleration. The infant’s gestational age was 41 weeks and 5 days, her birth weight was 3266 g, and her APGAR scores were 7 (1 min) and 8 (5 min). She showed no contractures in the upper and lower limbs.

Although she cried at birth, she was transferred to the neonatal intensive care unit of the hospital because of continued labored breathing. She also showed floppy status just after birth. The amniotic fluid was markedly clouded and her X-ray at the time showed patchy shadows throughout the lung field, compatible with meconium aspiration syndrome (MAS). Her respiration was assisted by positive airway pressure with oxygen for five days after birth, followed by spontaneous breathing with oxygen for five more days. Although the respiratory condition improved, muscle hypotonia was observed from 11 days of age. At day 20, systemic hypotonia and poor feeding were significant.

At 21 days of age, she was found to be SMA-NBS positive and was transferred to the Department of Pediatrics, Kobe University Hospital. At the time of admission, she was severely hypotonic with extremely weak muscles throughout her body. Anti-gravitational movement of the elbow joint was not observed. Her tendon reflexes were absent, but fasciculation in the tongue was not found. She showed seesaw breathing with a subcostal depression during inspiration. At 23 days of age, the results of genetic testing with MLPA and ddPCR were available (Figure 3): she carried homozygous deletion of *SMN1* exons 7 and 8 and two copies of *SMN2* exons 7 and 8. She also showed homozygous deletion of the neuronal apoptosis inhibitory protein (*NAIP*) gene exon 5 on MLPA analysis.

The patient was given nusinersen as the first emergency treatment at 25 days of age. The Children’s Hospital of Philadelphia Infant Test of Neuromuscular Disorders (CHOP-INTEND) score prior to the first administration was 12/64 (range, 0 to 64). Thereafter, nusinersen administration was carried out at 40, 55, and 90 days of age according to the initial regimen of loading dosing. The CHOP-INTEND score prior to the fourth administration (13 weeks of age) was 34/64 (Figure 3).

The parents of the patient decided to switch nusinersen to onasemnogene abeparvovec. Then, onasemnogene abeparvovec was administered at 4 months of age (19 weeks of age). The CHOP-INTEND scores were 43/64 prior to treatment with onasemnogene abeparvovec and 44/64 at one month after treatment. After 4 months of age, the acquisition rate of motor function slowed and the ability to suckle/swallow decreased. She showed no weight gain for 1.5 months, so tube feeding was started.

The compound muscle action potential (CMAP) amplitude of the median nerve at 33 days of age was 0.08 mV, showing a significant decrease.

### 3.6. Patient 2: A Pre-Symptomatic Patient Identified by SMA-NBS

Patient 2 was a three-month-old female infant diagnosed with pre-symptomatic SMA type I. She was born in June 2022. She was the first child of non-consanguineous healthy parents. The parents had no family histories of neuromuscular disorders. Her clinical information related to SMA diagnosis and treatments is summarized in Table 4 and Figure 2 and Figure 3.

During pregnancy, there were no notable episodes in the mother and infant. The patient was delivered from the mother who underwent epidural anesthesia. However, she did not show any respiratory problems. Her gestational age was 41 weeks and 2 days and her birth weight was 3120 g.

Five days after birth, the infant was discharged from the hospital with no specific problems. However, at 17 days of age, she was found to be SMA-NBS positive. The next day, she was referred to the Department of Pediatrics, Kobe University Hospital. At the time of admission, she showed brisk movements of the upper and lower limbs. Her tendon reflexes were normal and fasciculation in the tongue was not found. She showed no respiratory problems in the physical examination. At 19 days of age, the results of genetic testing with MLPA and ddPCR were available (Figure 2): she carried homozygous deletion of *SMN1* exons 7 and 8 and two copies of *SMN2* exons 7 and 8. She also showed homozygous deletion of *NAIP* exon 5 on MLPA analysis. Based on these findings, she was diagnosed with pre-symptomatic SMA type I.

The patient was given nusinersen as the first emergency treatment at 22 days of age. The CHOP-INTEND score prior to the first administration was 42/64. Thereafter, nusinersen administration was carried out at 36, 49, and 87 days of age according to the initial regimen of loading dosing. The CHOP-INTEND score prior to the fourth administration (12 weeks of age) was 44/64.

The CMAP amplitude of the ulnar nerve at 24 days of age was 3.5 mV, showing no decrease.

## 4. Discussion

### 4.1. Newborn Infants with SMA and Aborted Fetuses in the Period 2007–2016

The mean incidence of SMA in Hyogo Prefecture in the period of 2007–2016 was 3.88 per 100,000 live births, which is almost compatible with the previous results [4,22]. According to Verhaart et al., the median incidence of SMA in Europe in the period 2011–2015 was 11.9 per 100,000 live births [2]. Thus, the incidence of SMA in Japan is generally lower than in Europe.

Considering the number of aborted fetuses with SMA, the estimated incidence of SMA was 4.96 per 100,000 live births, increasing to 128% of the real incidence (3.88 to 4.96 in 100,000 live births). Our data suggested that one out of five fetuses with SMA was aborted based on the results of prenatal testing, and the remaining four fetuses were born and diagnosed with SMA because of motor dysfunction and respiratory failure (Table 2).

Until recently, some families with SMA have chosen abortion if prenatal testing revealed that the fetus had SMA. In the future, a positive prenatal testing result for SMA might not always lead to abortion of the affected fetuses, if the affected infants could be treated in the pre-symptomatic stage with effective drugs and expected to grow normally or sub-normally [14,15,16].

### 4.2. Incidence Rates of SMA Based on Newborn Screening

As SMA-NBS has spread more widely, the number of reports on the SMA incidence rate based on SMA-NBS has been increasing rapidly [23,24,25,26,27,28,29]. For both clinical-diagnosis-based incidence and genetic-diagnosis-based incidence, the number of births in vital statistics including non-examined infants is used as the parameter. In the SMA-NBS-based incidence, on the other hand, the number of newborn infants tested in the NBS program and confirmed by experts is used as the parameter. Therefore, the incidence rates based on SMA-NBS could be more accurate than those based on a clinical or genetic diagnosis. However, it is a prerequisite condition that all newborn infants participate in SMA-NBS.

In Hyogo Prefecture, the number of infants participating in SMA-NBS is only 8336 (which is ~22% of newborn infants), and two infants with SMA type I have already been identified by SMA-NBS (as of the end of August). At this moment, we cannot estimate the incidence rate of SMA based on these values. We have identified two barriers to the implementation of SMA-NBS: (1) parents are unaware of this serious but common genetic disease, and (2) parents do not know that SMA can be treated before onset. According to a recent report, many people in Hyogo Prefecture have a positive opinion of SMA-NBS, although their knowledge about SMA and its new treatments is very limited [30]. Thus, we believe that, if expectant parents are educated about SMA and its treatments, SMA-NBS will soon spread throughout Hyogo Prefecture, and then an accurate incidence rate will be able to be obtained.

### 4.3. False-Positive Cases in Hyogo Prefecture

There have been many reports that describe no false-positive cases with SMA-NBS [19,26,31,32,33]. D’Silva et al. screened 252,081 newborn infants with a false-positive rate of less than 0.001% in Newss South Wales [34]. In general, false-positive results are unlikely to occur in SMA-NBS, despite differences in methodology. Even so, there are some reports that describe the causes of false-positive results in the SMA-NBS [23,35,36]. The causes of false-positive results may be divided into three groups: genetic variation of *SMN1* [23], DNA quality and/or quantity of the DBS samples [35,36], and instrument performance in detecting *SMN1* gene deletion [36].

In Hyogo Prefecture, we screened 8336 newborn infants and found ten false-positive cases during the study period from February 2021 to August 2022 (false-positive rate: 0.12%) (Table 3). Most of our false-positive cases were born in the core hospitals in Hyogo Prefecture. These core hospitals accept many high-risk infants. According to our data, four out of the ten false-positive samples were from high-risk infants.

For the treatment or management of high-risk infants, catheters are often placed in the umbilical cord or other blood vessels. These catheters are placed for the frequent injection of drugs (e.g., antibiotics) or frequent monitoring (e.g., blood gases). Blood is also drawn from umbilical catheters, arterial catheters, and central or peripheral venous catheters. In these cases, to prevent catheter occlusion, normal saline (0.9% sodium chloride) is continuously infused or heparinized saline is intermittently flushed [37]. In such cases, diluted or heparinized blood can be used for the preparation of DBS samples. In addition, there remains a possibility that, in these core hospitals, heel-prick blood was collected using a heparin-coated capillary when DBS samples were prepared for SMA-NBS.

Therefore, we hypothesized that heparinization and/or dilution of the blood might be related to the high frequency of false-positive cases in Hyogo Prefecture. Our experiments strongly suggest that our hypothesis may be correct. We have recently requested again that obstetricians and neonatologists do not use heparinized blood when preparing SMA-NBS samples.

### 4.4. SMA-NBS and Pre-Symptomatic Treatment

Now, the SMA-NBS program is recognized as essential for enabling patient treatment in the pre-symptomatic stage [17,38,39]. The NURTURE trial of nusinersen demonstrated that the outcomes of treatment in pre-symptomatic infants with two or three copies of *SMN2* were much more advantageous compared with treatment in symptomatic patients [14]. The SPR1NT trial of onasemnogene abeparvovec also demonstrated that neonates treated pre-symptomatically achieved greater and earlier developmental milestones than both untreated patients and patients who were not treated until they were in the symptomatic stage [15,16]. We now have reached the understanding that SMA should be treated in the pre-symptomatic stage.

In our study, two screen-positive cases were diagnosed with SMA type I by confirmatory genetic testing. They carried homozygous deletion of *SMN1* exons 7 and 8 and two copies of *SMN2* exons 7 and 8. They also showed homozygous deletion of *NAIP* exon 5 on MLPA analysis. These combined results of *SMN2* copy number and *NAIP* deletion are often observed in patients with SMA type I [40].

One of them, Patient 2, had not shown any signs and symptoms at the time of diagnosis and initiation of treatment. In this situation, pre-symptomatic treatment with nusinersen was carried out. However, the family found the diagnosis of SMA harder to accept because the infant appeared completely healthy. We repeatedly explained to the parents the benefits of early treatment, and the parents eventually consented to treatment for the baby.

Underlying these behaviors may be fear for the future of their baby, a lack of confidence in taking appropriate action for their baby, and concern about possible social exclusion of the baby and family (stigmatization) [41]. Families were dominated by such negative feelings rather than feeling gratitude for the early diagnosis of the disease. In such cases, it is essential that not only treatment of the child but also psychological support for the family is available.

### 4.5. SMA-NBS and Post-Symptomatic Treatment

Regarding the other patient (Patient 1) in our study, symptoms and signs appeared before the screening results were obtained. The patient suffered from respiratory failure and was managed under a diagnosis of MAS. The meconium-stained amniotic fluid and chest X-ray findings were compatible with MAS, so SMA might unfortunately not have been considered as a cause of respiratory failure. The patient may have been symptomatic at birth, and the diagnosis of SMA might have been delayed because of the presence of MAS, which required management by the obstetricians and neonatologists. Even so, the patient was identified by SMA-NBS and treated with nusinersen at 25 days of age. She has made modest motor improvements since treatment initiation with a 32-point improvement in the CHOP-INTEND score.

Patient 1, who had two copies of *SMN2*, should have been diagnosed as SMA type I-A. In some SMA infants, the symptoms may be present at birth (SMA type 0) or within the first 2 weeks of life (SMA type I-A: the most severe form of SMA type I) [42]. Patients with SMA type 0 and type I-A may have a single copy and two copies of *SMN2*, respectively [43]. However, considering the continuous spectrum of phenotypes in SMA, it would be difficult to differentiate between type 0 and type I-A disease, and both categories could be merged into type 0/I-A disease [44]. Kitaoka et al. reported an SMA infant with two copies of *SMN2* as a type 0 case, because of early onset at birth and claw-like hand deformities [45].

In Patient 1, SMA-NBS was unable to lead to pre-symptomatic treatment. There is an article entitled “Spinal Muscular Atrophy—Is Newborn Screening Too Late for Children with Two *SMN2* Copies?” [46]. According to the authors of this article, more than 70% of SMA patients with two *SMN2* copies can achieve independent ambulation with immediate initiation of therapy, but they simultaneously suggested the possibility of less favorable outcomes in the remainder of patients [46]. However, in the case of Patient 1, without SMA-NBS, diagnosis and treatment might have been further delayed. Further delayed diagnosis might have resulted in a much less favorable outcome, potentially infantile death.

### 4.6. Research Limitations

First, the estimates of SMA incidence in Table 2 were based on the number of symptomatic SMA patients diagnosed by genetic testing performed at Kobe University. Most SMA patients in Hyogo Prefecture were diagnosed at Kobe University, but it remains possible that a small number of patients were diagnosed at other institutions.

Second, the number of infants participating in SMA-NBS was too small in this study, which could not lead to an estimation of the incidence of SMA.

## 5. Conclusions

In Hyogo Prefecture, Japan, we performed a pilot study of SMA-NBS between February 2021 and August 2022. Hyogo Prefecture has ~40,000 live births per year, and the estimated incidence of SMA was 1 in 20,000–25,000 based on the genetic testing of symptomatic patients with SMA.

Of the 8336 newborns screened in this study, 12 screen-positive cases were detected by the real-time PCR assay. The MLPA assay excluded ten false positives and identified two patients. The false positives might be related to heparinized blood in the DBS sample. Both patients carried two copies of *SMN2*; one was asymptomatic and the other was symptomatic at the time of diagnosis.

SMA-NBS does not guarantee pre-symptomatic treatment. However, SMA-NBS enables us to prevent delayed diagnosis of SMA, even if it does not always lead to treatment in the pre-symptomatic stage. It allows us to provide symptomatic patients with early diagnosis and treatment to ensure the best possible outcome.

## Figures and Tables

**Figure 1 genes-13-02110-f001:**
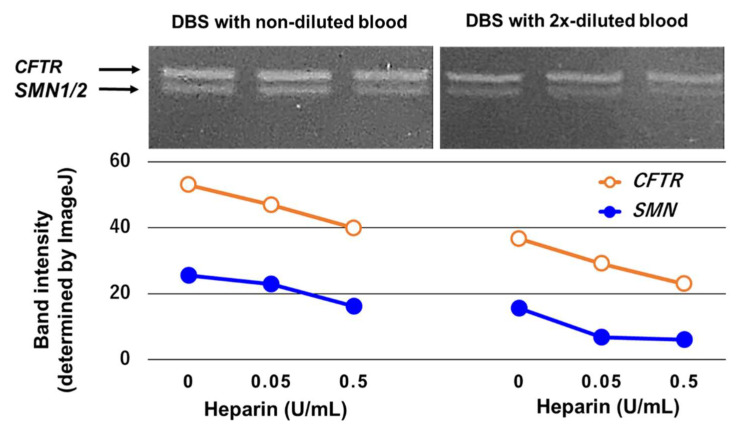
Influencing factors for PCR amplification. We tested the effects of white blood cell count, gene sequence, and heparin on the PCR amplification efficiency (Appendix A). In this experiment, we obtained three findings. First, the PCR product amounts of two-times diluted samples (half of the normal blood cell count) were less than those of non-diluted samples (normal blood cell count). Second, the PCR product amounts of *SMN* were much smaller than those of *CFTR* (a non-*SMN* gene). Third, the PCR product amounts of *SMN* and *CFTR* decreased in the presence of heparin. The decrease was in a heparin dose-dependent manner. *SMN*: survival motor neuron. *CFTR*: cystic fibrosis transmembrane conductance regulator.

**Figure 2 genes-13-02110-f002:**
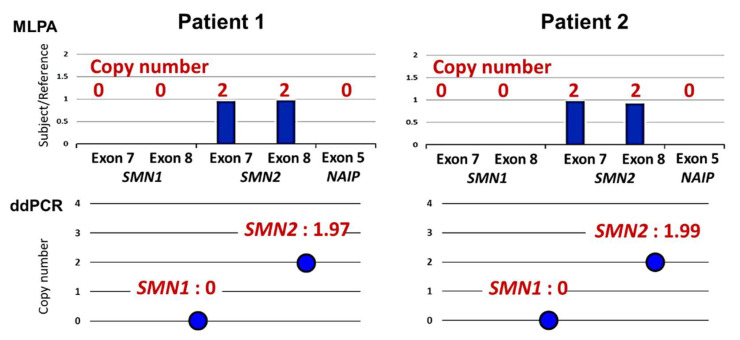
MLPA and ddPCR analyses of patients identified by SMA-NBS. Multiplex ligation-dependent probe amplification (MLPA) showed that both patients carried homozygous deletion of *SMN1* exons 7 and 8 and two copies of *SMN2* exons 7 and 8. They also showed homozygous deletion of *NAIP* exon 5 on MLPA analysis. *NAIP*: the neuronal apoptosis inhibitory protein gene. Droplet digital PCR (ddPCR) showed that both patients carried homozygous deletion of *SMN1* exons 7 and two copies of *SMN2* exons 7, which was consistent with the results of MLPA analysis.

**Figure 3 genes-13-02110-f003:**
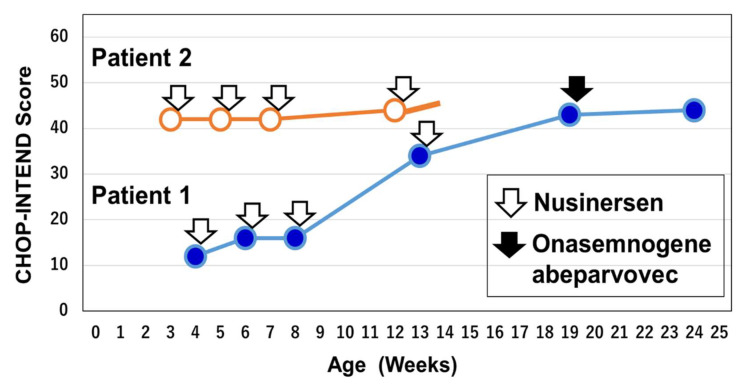
Clinical courses of Patients 1 and 2. Patient 1 may have been symptomatic at birth and diagnosis of SMA might have been delayed. However, the patient was treated with nusinersen at 24 days of age. Then, onasemnogene abeparvovec was administered at 4 months of age (19 weeks of age). The patient has made modest motor improvements since treatment initiation with a 32-point improvement in CHOP-INTEND score (12-point before treatment to 44-point at 24 weeks). Patient 2 was treated pre-symptomatically. These patients remain under observation. CHOP-INTEND: The Children’s Hospital of Philadelphia Infant Test of Neuromuscular Disorders.

**Table 1 genes-13-02110-t001:** Studies related to SMA-NBS in Hyogo Prefecture.

**(1) Prediction of the number of SMA patients**
Incidence studies of SMA before SMA-NBS started (Section 3.1 and Section 3.2)
**(2) Pitfalls of SMA-NBS**
Case studies with false-positive results (Section 3.3 and Section 3.4)
**(3) Patients identified by SMA-NBS**
Case reports of patients received pre-symptomatic or post-symptomatic treatment (Section 3.5 and Section 3.6)

SMA-NBS: newborn screening for spinal muscular atrophy.

**Table 2 genes-13-02110-t002:** Incidence of SMA patients in Hyogo Prefecture.

		Number of Patients with SMA Types I, II, III			Number of Patients with SMA Type I		
	Live Births	Affected Infants	Affected Fetuses	Affected Infants and Fetuses	Affected Infants	Affected Fetus	Affected Infants and Fetuses
**2007**	48,685	2	1	3	1	1	2
**2008**	48,833	3	0	3	2	0	2
**2009**	47,592	1	0	1	0	0	0
**2010**	47,834	2	0	2	1	0	1
**2011**	47,351	1	0	1	1	0	1
**2012**	46,436	1	0	1	1	0	1
**2013**	45,673	3	0	3	1	0	1
**2014**	44,352	3	3	6	1	2	3
**2015**	44,015	1	0	1	0	0	0
**2016**	43,378	1	1	2	1	1	2
**Total**	464,149	18	5	23	9	4	13
**Average**	46,415	1.8	0.5	2.3	0.9	0.4	1.3
**Incidence (in 100,000)**		Affected infants only 3.88 [95% CI: 0.02, 7.74]			Affected infants only 1.94 [95%CI: −0.79, 4.67]		
		Affected infants and fetuses 4.96 [95% CI: 0.59, 9.32]			Affected infants and fetuses 2.80 [95%CI: −0.48, 6.08]		

**Table 3 genes-13-02110-t003:** False-positive cases.

Case	1	2	3	4	5	6	7	8	9	10
**Hospital**	A	B	B	B	B	B	C	B	B	B
**Year** **(month)**	2021 (March)	2021 (August)	2021 (August)	2021 (August)	2021 (August)	2021 (September)	2021 (November)	2022 (May)	2022 (July)	2022 (July)
**Sex**	male	female	male	male	female	male	female	male	male	male
**Gestational age**	37 weeks	40 weeks	34 weeks	36 weeks	39 weeks	38 weeks	31 weeks	39 weeks	40 weeks	41 weeks
**Birth weight**	2188 g	3084 g	2840 g	2572 g	2620 g	3800 g	1588 g	3990 g	3372 g	3478 g
**Events in perinatal period**	gastric mucosal lesion	none	none	apnea	none	pneumo-thorax	low birth weight	none	none	none
***SMN1* copy number ***	2	2	2	2	2	2	2	2	2	2

A, B, and C are core hospitals in the west area, the central area, and the north area of Hyogo Prefecture. * *SMN1* copies were determined by multiplex ligation-dependent probe amplification (MLPA).

**Table 4 genes-13-02110-t004:** Summary of the patients.

	Patient 1	Patient 2
**Sex**	female	female
**Weight**	3266 g	3120 g
**Gestational age**	41 weeks and 5 days	41 weeks and 2 days
**DBS sampling**	4 days	4 days
**Notification of NBS results**	21 days	17 days
**Examination and inspection**	22 days	18 days
**Notification of inspection results**	24 days	21 days
**Events in peri-neonatal period**	MAS and floppy status	healthy condition
**Initiation of SMA treatment** **(First injection of nusinersen)**	25 days	22 days
**Gene therapy with** **onasemnogene abeparvovec**	19 weeks	not done
**CHOP-INTEND score**	(23 days; before treatment) 12 (13 weeks) 34 (24 weeks) 44	(21 days; before treatment) 42 (12 weeks) 44
**CMAP amplitude**	(33 days, median nerve) 0.08 mV	(24 days, ulnar nerve) 3.5 mV

CHOP-INTEND: The Children’s Hospital of Philadelphia Infant Test of Neuromuscular Disorders. CMAP: compound muscle action potential. MAS: meconium aspiration syndrome.

## Data Availability

Not applicable. The data that support the findings of this study are available from the corresponding author, H.N., upon reasonable request.

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
