# Peer review of "PCR-Based Screening of Spinal Muscular Atrophy for Newborn Infants in Hyogo Prefecture, Japan"

_genes, 2022, doi:10.3390/genes13112110_

Round 1

Reviewer 1 Report

This manuscript describes a case study for the PCR screening for spinal muscular atrophy in newborn infants in Japan. The methods are rigorous described in adequate detail and the  data presentation is clear. Upon review I recommend this article for publication.

Reviewer 2 Report

The mansucript of Noguchi et al. is presenting the first results of PCR-based Screening for SMA in one of the Japanese prefecture between February 2021 and August 2022. An unexpected fouding is a large number of high false positve cases, however this phenomenon is well diucussed by the authors. Also, the disease course of two patients is decribed in detail.

The topic is of interest to the community working on genetics and pediatrics. The mansucript is very logically written, clear and understandable. The methodology and results are sound and well decribed. The conculsions are structured, formulated in three clear points. 

Altogether I found the manuscript well-written and structured. The topic is interesting to the general public given the therapeutic possibility. The positive numbers are quite small, but it is understandable given the duration of the program and the fact that data are only from one prefecture. The introduction is providing a structured background for the one that does not have much orientation in the topic. Methods are clear and the tables are facilitating the understanding of text. Results are also presented clearly and soundly from the scientific point of view. The discussion encompasses the most important points, but it is concise. The conclusions are written into three points.   Regarding the scientific part, I agree with the authors that an exact incidence rate is difficult to obtain based on the available data. The two patients are well characterized. The most interesting finding I found was a high false positives rate and the discussion on the influence of heparinized/diluted blood. The hypothesis is well illustrated with images.   From my point of view, the manuscript can be accepted in its current form. However, if I were to give some comments or questions they would be:   1. At what stage was genetic counseling provided and by whom? Did parents of the false positives children get genetic counseling as well? 2. Is any data from other prefectures in Japan published on this topic? 3. Can there be any other explanation for the high false positives (apart from the dilution/heparinization) theory? Are the same reagents as in other countries used? Are there differences with test interpretation between countries?